# PROSPECT PRUNING: FINDING TRAINABLE WEIGHTS AT INITIALIZATION USING META-GRADIENTS

**Milad Alizadeh**[*1,2]  **Shyam A. Tailor**[2]  **Luisa Zintgraf**[3]  **Joost van Amersfoort**[1]
**Sebastian Farquhar**[1]  **Nicholas Donald Lane**[2,4]  **Yarin Gal**[1]

[1]OATML, Department of Computer Science, University of Oxford
[2]CaMLSys, Department of Computer Science and Technology, University of Cambridge
[3]WhiRL, Department of Computer Science, University of Oxford
[4]Samsung AI Center, Cambridge

## ABSTRACT

Pruning neural networks at initialization would enable us to find sparse models that retain the accuracy of the original network while consuming fewer computational resources for training and inference. However, current methods are insufficient to enable this optimization and lead to a large degradation in model performance. In this paper, we identify a fundamental limitation in the formulation of current methods, namely that their saliency criteria look at a single step at the start of training without taking into account the *trainability* of the network. While pruning iteratively and gradually has been shown to improve pruning performance, explicit consideration of the training stage that will immediately follow pruning has so far been absent from the computation of the saliency criterion. To overcome the short-sightedness of existing methods, we propose Prospect Pruning (ProsPr), which uses *meta-gradients* through the *first few steps of optimization* to determine which weights to prune. ProsPr combines an estimate of the higher-order effects of pruning on the loss and the optimization trajectory to identify the trainable sub-network. Our method achieves state-of-the-art pruning performance on a variety of vision classification tasks, with less data and in a single shot compared to existing pruning-at-initialization methods. Our code is available online at `https://github.com/mil-ad/prospr`.

## 1 INTRODUCTION

Pruning at *initialization*—where we remove weights from a model before training begins—is a recent and promising area of research that enables us to enjoy the benefits of pruning at training time, and which may aid our understanding of training deep neural networks.

Frankle & Carbin (2019) provide empirical evidence for the existence of sparse sub-networks that can be trained from initialization and achieve accuracies comparable to the original network. These "winning tickets" were originally found in an iterative process where, in each iteration, the network is trained to full convergence followed by pruning a subset of the weights by magnitude. The values of the remaining weights are then rewound to their value at initialization, and the process is repeated iteratively until the desired sparsity level is achieved.

This process, known as Lottery Ticket Rewinding (LTR), is very compute-intensive and is prone to failures. For instance, Frankle et al. (2020) show better results by rewinding weights not all the way back to initialization, but to early stages of training instead. LTR is especially prone to failure for more difficult problems (e.g., training on ImageNet), where we must rewind weights to their state several epochs into training.

A recent line of work proposes alternative practical solutions to identify these sub-networks *before* training begins, without the cost of retraining the network iteratively Lee et al. (2018); Wang et al. (2020); de Jorge et al. (2021); Tanaka et al. (2020). This class of methods uses gradients to assess

---

[*]Corresponding author. Contact at `milad.alizadeh@cs.ox.ac.uk`

the importance of neural network weights. These gradients are often known as Synaptic Saliencies and are used to estimate the effect of pruning a *single* parameter in isolation on various objectives, typically the loss function. This objective is not so different from classical pruning-at-convergence methods, but the gradients for a well-trained model are small; therefore these methods must inspect higher-order metrics such as the Hessian to estimate the pruning effect (LeCun et al., 1990; Hassibi & Stork, 1993). Pruning at initialization is desirable because the benefits of pruning (in terms of memory and speed) can be reaped during training, rather than only at inference/deployment time.

However, the performance of prune-at-init methods remains poor: the degradation in accuracy is still significant compared to training the full model and LTR, making these methods impractical for many real-world problems (Frankle et al., 2021). In this paper, we identify a fundamental limitation in the objective formulation of current methods, namely that saliency criteria do not take into account the fact that the model is going to be trained after the pruning step. If our aim was to simply prune a subset of weights without affecting the loss, then these saliency criteria are estimating the correct objective. However, this estimate does not take into account that we are going to train the weights after we prune them. We need a metric that captures the *trainability* of the weights during the optimization steps, rather than a single myopic estimate.

Many methods attempt to overcome this by pruning gradually and/or adding training steps between iterative pruning steps (Zhu & Gupta, 2018; You et al., 2020; de Jorge et al., 2021). Although this approach has been shown to be effective, it is expensive and cumbersome in practice and ultimately is an indirect approximation to the *trainability* criteria we are looking to incorporate into our objective.

In this paper, we propose Prospect Pruning (**ProsPr**), a new pruning-at-init method that learns from the first few steps of optimization which parameters to prune. We explicitly formulate our saliency criteria to account for the fact that the network will be trained after pruning. More precisely, ProsPr uses *meta-gradients* by backpropagating through the first few model updates in order to estimate the effect the initial pruning parameters have on the loss after a few gradient descent steps. Effectively this enables us to account for both higher-order effects of pruning weights on the loss, as well as the trainability of individual weights. Similar to other methods we apply pruning to initialization values of weights and train our models from scratch. In summary, our contributions are:

- We identify a key limitation in prior saliency criteria for pruning neural networks—namely that they do not explicitly incorporate trainability-after-pruning into their criteria.
- We propose a new pruning-at-init method, **ProsPr**, that uses meta-gradients over the first few training steps to bridge the gap between pruning and training.
- We show empirically that ProsPr achieves higher accuracy compared to existing pruning-at-init methods. Unlike other methods, our approach is *single shot* in the sense that the pruning is applied to the network initial weights in a single step.

## 2 BACKGROUND

In this section we review the key concepts that our method builds upon. We delay comparisons to other pruning techniques in the literature to Section 5.

Classic post-training pruning methods aim to identify and remove network weights with the least impact on the loss (LeCun et al., 1990; Hassibi & Stork, 1993). They typically use the Taylor expansion of the loss with respect to parameters to define a saliency score for each parameter: $\delta\mathcal{L} \approx \nabla_{\boldsymbol{\theta}}\mathcal{L}^{\top}\delta\boldsymbol{\theta} + \frac{1}{2}\delta\boldsymbol{\theta}^{\top}\mathbf{H}\,\delta\boldsymbol{\theta}$, where $\mathbf{H} = \nabla_{\boldsymbol{\theta}}^2\mathcal{L}$ is the Hessian matrix. When the network has converged, the first-order term in the expansion is negligible, and hence these methods resort to using $\mathbf{H}$.

Lee et al. (2018) introduce SNIP, and show that the same objective of minimizing the change in loss can be used *at initialization* to obtain a trainable pruned network. At initialization, the first-order gradients $\nabla_{\boldsymbol{\theta}}$ in the local quadratic approximation are still significant, so higher-order terms can be ignored. Hence the computation of the parameter saliencies can be done using backpropagation.

The Taylor expansion approximates the effect of small *additive* perturbations to the loss. To better approximate the effect of *removing* a weight, Lee et al. (2018) attach a *multiplicative* all-one mask to the computation graph of each weight. This does not change the forward-pass of the network, but it enables us to form the Taylor expansion around the mask values, rather than the weights, to estimate the effect of changing the mask values from 1 to 0. More specifically, SNIP computes the

saliency scores according to:

$$s_j = \frac{|g_j(\mathbf{w}, \mathcal{D})|}{\sum_{k=1}^m |g_k(\mathbf{w}, \mathcal{D})|} \; , \tag{1}$$

with

$$g_j(\mathbf{w}, \mathcal{D}) = \frac{\partial \mathcal{L}(\mathbf{c} \odot \mathbf{w}, \mathcal{D})}{\partial c_j} \; , \tag{2}$$

where $m$ is the number of weights in the network, $\mathbf{c} \in \{0, 1\}^m$ is the pruning mask (initialised to $\mathbf{1}$ above), $\mathcal{D}$ is the training dataset, $\mathbf{w}$ are the neural network weights, $\mathcal{L}$ is the loss function, and $\odot$ is the Hadamard product. These saliency scores are computed before training the network, using one (or more) mini-batches from the training set. The global Top-K weights with the highest saliency scores are retained ($c_j = 1$), and all other weights are pruned ($c_j = 0$), before the network is trained.

Our method, to be introduced in Section 3, also relies on computing the saliency scores for each mask element, but uses a more sophisticated loss function to incorporate the notion of trainability.

## 3    OUR METHOD: PROSPR

In this section we introduce our method, Prospect Pruning (ProsPr). We note that for the problem of pruning at initialization, the pruning step is immediately followed by training. Therefore, pruning should take into account the *trainability* of a weight, instead of only its immediate impact on the loss before training. In other words, we want to be able to identify weights that are not only important at initialization, but which may be useful for reducing the loss *during training*. To this end, we propose to estimate the effect of pruning on the loss over *several steps of gradient descent* at the beginning of training, rather than the changes in loss at initialization.

More specifically, ProsPr models how training would happen by performing multiple (M) iterations of backpropagation and weight updates—like during normal training. We can then backpropagate through the entire computation graph, from the loss several steps into training, back to the original mask, since the gradient descent procedure is itself differentiable. Once the pruning mask is computed, we rewind the weights back to their values at initialization and train the pruned network. The gradient-of-gradients is called a *meta-gradient*. This algorithm is illustrated visually in Figure 1.

The higher-order information in the meta-gradient includes interactions between the weights during training. When pruning at initialization, our ultimate goal is to pick a pruned model, A, which is more trainable than an alternative pruned model B. That means we want the loss $\mathcal{L}(\hat{y}_A, y)$ to be lower than $\mathcal{L}(\hat{y}_B, y)$ at convergence (for a fixed pruning ratio). Finding the optimal pruning mask is generally infeasible since the training horizon is long (i.e., evaluation is costly) and the space of possible pruning masks is large. Unlike other methods that must compute the saliency scores iteratively, we can use the meta-gradients to compute the pruning mask in *one shot*. This picks a line in loss-space, which more closely predicts the eventual actual loss. This is because it smooths out over more steps, and takes into account interactions between weights in the training dynamics. Crucially, in the limit of large M, the match to the ultimate objective is exact.

### 3.1    SALIENCY SCORES VIA META-GRADIENTS

We now introduce ProsPr formally. After initialising the network weights randomly to obtain $\mathbf{w}_{\text{init}}$, we apply a weight mask to the initial weights,

$$\mathbf{w}_0 = \mathbf{c} \odot \mathbf{w}_{\text{init}}. \tag{3}$$

This weight mask contains only ones, $\mathbf{c} = \mathbf{1}$, as in SNIP (Lee et al., 2018), and represents the connectivity of the corresponding weights.

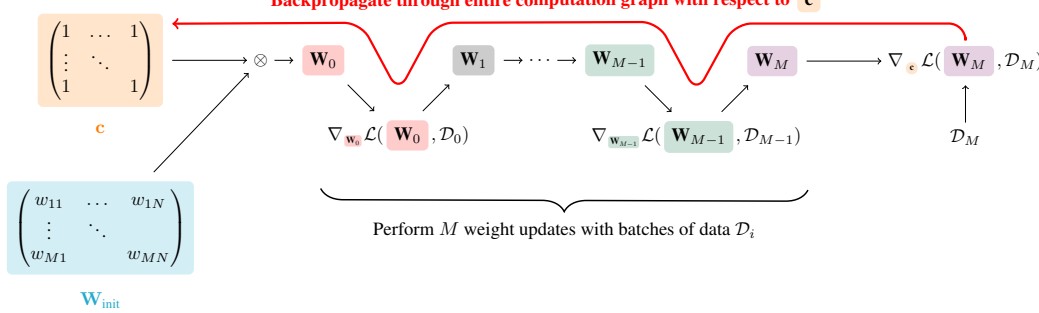

Figure 1: Visualization of computing saliency scores in our method, ProsPr. By backpropagating through several gradient steps we capture higher-order information about the objective that we care about in practice, i.e., saliency of parameters *during training* and not just at initialization.

We then sample $M+1$ batches of data $\mathcal{D}_i \sim \mathcal{D}^{\text{train}}$ ($i \in \{0, \dots, M\}$; $M \geq 1$) for the pruning step, and perform $M$ weight updates[1],

$$\mathbf{w}_1 = \mathbf{w}_0 - \alpha \nabla_{\mathbf{w}_0} \mathcal{L}(\mathbf{w}_0, \mathcal{D}_0) \tag{4}$$

$$\vdots$$

$$\mathbf{w}_M = \mathbf{w}_{M-1} - \alpha \nabla_{\mathbf{w}_{M-1}} \mathcal{L}(\mathbf{w}_{M-1}, \mathcal{D}_{M-1}). \tag{5}$$

Then, we compute a meta-gradient that backpropagates through these updates. Specifically, we compute the gradient of the final loss w.r.t. the initial mask,

$$\nabla_{\mathbf{c}} \mathcal{L}(\mathbf{w}_M, \mathcal{D}_M). \tag{6}$$

Using the chain rule, we can write out the form of the meta-gradient beginning from the last step:

$$\nabla_{\mathbf{c}} \mathcal{L}(\mathbf{w}_M, \mathcal{D}) = \nabla_{\mathbf{w}_M} \mathcal{L}(\mathbf{w}_M, \mathcal{D})(\nabla_{\mathbf{c}} \mathbf{w}_M), \tag{7}$$

repeating for each step until we reach the zero'th step whose gradient is trivial,

$$= \nabla_{\mathbf{w}_M} \mathcal{L}(\mathbf{w}_M, \mathcal{D})(\nabla_{\mathbf{w}_{M-1}} \mathbf{w}_M) \dots (\nabla_{\mathbf{w}_0} \mathbf{w}_1)(\nabla_c \mathbf{w}_0) \tag{8}$$

$$= \nabla_{\mathbf{w}_M} \mathcal{L}(\mathbf{w}_M, \mathcal{D})(\nabla_{\mathbf{w}_{M-1}} \mathbf{w}_M) \dots (\nabla_{\mathbf{w}_0} \mathbf{w}_1)(\nabla_c(\mathbf{c} \odot \mathbf{w}_{\text{init}})) \tag{9}$$

$$= \nabla_{\mathbf{w}_M} \mathcal{L}(\mathbf{w}_M, \mathcal{D}) \left[ \prod_{m=1}^{M} (\nabla_{\mathbf{w}_{m-1}} \mathbf{w}_m) \right] \mathbf{w}_{\text{init}}. \tag{10}$$

In practice, we can compute the meta-gradients by relying on automatic differentiation software such as PyTorch (Paszke et al., 2019). However, care must be taken to ensure that weights at each step are kept in memory so that the entire computation graph, including gradients, is visible to the automatic differentiation software. The saliency scores are now given by

$$s_j = \frac{|g_j(\mathbf{w}, \mathcal{D})|}{\sum_{k=1}^{m} |g_k(\mathbf{w}, \mathcal{D})|}, \tag{11}$$

with

$$g_j(\mathbf{w}, \mathcal{D}) = \frac{\partial \mathcal{L}(\mathbf{w}_M, \mathcal{D})}{\partial c_j}, \tag{12}$$

where $\mathbf{w}_M$ is a function of $\mathbf{c}$. Equation (12) stands in contrast to SNIP, where the saliency is computed using the loss at $\mathbf{c} \cdot \mathbf{w}_{\text{init}}$ rather than $\mathbf{w}_M$. The saliency scores are then used to prune the *initial* weights $\mathbf{w}_{\text{init}}$: the ones with the highest saliency scores are retained ($c_j = 1$), and all other weights are pruned ($c_j = 0$). Finally, the network is trained with the pruned weights $\hat{\mathbf{w}}_{\text{init}}$.

Algorithm 1 summarises the proposed method, ProsPr.

---

[1]We formalise the weight updates using vanilla SGD here; in practice these may be different when using approaches such as momentum or BatchNorm (Ioffe & Szegedy, 2015). Since our implementation relies on automatic differentiation in PyTorch (Paszke et al., 2019), we can use any type of update, as long as it is differentiable w.r.t. the initial mask $\mathbf{c}$.

---

**Algorithm 1** ProsPr Pseudo-Code

---

1: Inputs: a training dataset $\mathcal{D}^{\text{train}}$, number of initial training steps $M$, number of main training steps $N$ ($M \ll N$), learning rate $\alpha$
2: Initialise: network weights $\mathbf{w}_{\text{init}}$

---

3: $\mathbf{c}_{\text{init}} = \mathbf{1}$        ▷ Initialise mask with ones
4: $\mathbf{w}_0 = \mathbf{c}_{\text{init}} \odot \mathbf{w}_{\text{init}}$        ▷ Apply mask to initial weights
5: **for** $k = 0, \ldots, M-1$ **do**
6:      $\mathcal{D}_k \sim \mathcal{D}^{\text{train}}$        ▷ Sample batch of data
7:      $\mathbf{w}_{i+1} = \mathbf{w}_i - \alpha \nabla_{\mathbf{w}} \mathcal{L}(\mathbf{w}_i, \mathcal{D}_k)$        ▷ Update network weights
8: **end for**

---

9: $g_j(\mathbf{w}, \mathcal{D}) = \partial \mathcal{L}(\mathbf{w}_M, \mathcal{D}) / \partial c_j$        ▷ Compute meta-gradient
10: $s_j = \frac{|g_j(\mathbf{w}, \mathcal{D})|}{\sum_{k=1}^{m} |g_k(\mathbf{w}, \mathcal{D})|}$        ▷ Compute saliency scores
11: Determine the k-th largest element in $\mathbf{s}$, $s_k$.
12: $\mathbf{c}_{\text{prune}} = \begin{cases} 1, & \text{if } c_j \geq s_k \\ 0, & \text{otherwise} \end{cases}$        ▷ Set pruning mask
13: $\hat{\mathbf{w}}_0 = \mathbf{c}_{\text{prune}} \odot \mathbf{w}_{\text{init}}$        ▷ Apply mask to initial weights $\mathbf{w}_{\text{init}}$

---

14: **for** $i = 1, \ldots, N$ **do**        ▷ Train pruned model
15:      $\hat{\mathbf{w}}_{i+1} = \hat{\mathbf{w}}_i - \alpha \nabla_{\mathbf{w}} \mathcal{L}(\hat{\mathbf{w}}_i, \mathcal{D})$
16: **end for**

---

## 3.2 First-Order Approximation

Taking the meta-gradient through many model updates (Equation (6)) can be memory intensive: in the forward pass, all gradients of the individual update steps need to be retained in memory to then be able to backpropagate all the way to the initial mask. However, we only need to perform a few steps[2] at the beginning of training so in practice we can perform the pruning step on CPU which usually has access to more memory compared to a GPU. We apply this approach in our own experiments, with overheads of around 30 seconds being observed for the pruning step.

Alternatively, when the number of training steps needs to be large we can use the following first-order approximation. Using Equation (10), the meta-gradient is:

$$\nabla_{\mathbf{c}} \mathcal{L}(\mathbf{w}_M, \mathcal{D}_M) = \nabla_{\mathbf{w}_M} \mathcal{L}(\mathbf{w}_M, \mathcal{D}_M) \left[ \prod_{m=1}^{M} \left( \nabla_{\mathbf{w}_{m-1}} \mathbf{w}_m \right) \right] \mathbf{w}_{\text{init}}, \tag{13}$$

writing $\mathbf{w}_m$ in terms of $\mathbf{w}_{m-1}$ following SGD,

$$= \nabla_{\mathbf{w}_M} \mathcal{L}(\mathbf{w}_M, \mathcal{D}_M) \left[ \prod_{m=1}^{M} \nabla_{\mathbf{w}_{m-1}} (\mathbf{w}_{m-1} - \alpha \nabla_{\mathbf{w}_{m-1}} \mathcal{L}(\mathbf{w}_{m-1}; \mathcal{D}_m)) \right] \mathbf{w}_{\text{init}}, \tag{14}$$

carrying through the partial derivative,

$$= \nabla_{\mathbf{w}_M} \mathcal{L}(\mathbf{w}_M, \mathcal{D}_M) \left[ \prod_{m=1}^{M} I - \alpha \nabla_{\mathbf{w}_{m-1}}^2 \mathcal{L}(\mathbf{w}_{m-1}; \mathcal{D}_m) \right] \mathbf{w}_{\text{init}}, \tag{15}$$

and finally dropping small terms for sufficiently small learning rates,

$$\approx \nabla_{\mathbf{w}_M} \mathcal{L}(\mathbf{w}_M, \mathcal{D}_M) \left[ \prod_{m=1}^{M} I \right] \mathbf{w}_{\text{init}}, \tag{16}$$

$$= \nabla_{\mathbf{w}_M} \mathcal{L}(\mathbf{w}_M, \mathcal{D}_M) \, \mathbf{w}_{\text{init}}. \tag{17}$$

In the second-to-last step, we drop the higher-order terms, which gives us a first-order approximation of the meta-gradient[3].

---

[2] We use 3 steps for experiments on CIFAR-10, CIFAR-100 and TinyImageNet datasets
[3] Note that this approximation also works for optimisers other than vanilla SGD (e.g., Adam, Adamw, Adabound), except that the term which is dropped (r.h.s. of Equation Equation (15)) looks slightly different.

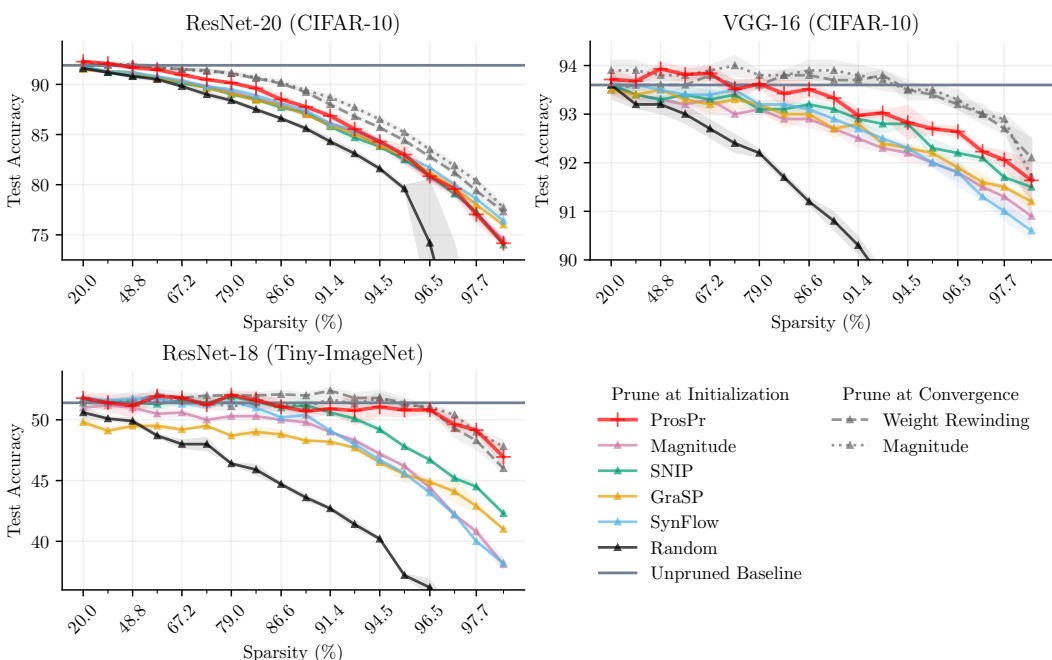

Figure 2: Accuracy of ProsPr against other prune-at-init and prune-after-convergence methods as benchmarked by Frankle et al. (2021). The shaded areas denote the standard deviation of the runs.

With this approximation, we only need to save the initial weight vector $\mathbf{w}_{\text{init}}$ in memory and multiply it with the final gradient. This approximation can be crude when the Laplacian terms are large, but with a sufficiently small learning rate it becomes precise. The approximation allows us to take many more intermediate gradient-steps which can be beneficial for performance when the training dataset has many classes, as we will see in Section 4.2.

## 4  EXPERIMENTS

We empirically evaluate the performance of our method, ProsPr, compared to various vision classification baselines across different architectures and datasets. In supplementary sections we show effectiveness of our method on image segmentation tasks (Appendix D) and when using self-supervised initialization (Appendix E). We provide details of our hyper-parameters, experiment setup, and implementation details in Appendix A.

### 4.1  RESULTS ON CIFAR AND TINY-IMAGENET

In recent work, Frankle et al. (2021) extensively study and evaluate different pruning-at-initialization methods under various effects such as weight re-initialization, weight shuffling, and score inversion. They report the best achievable results by these methods and highlight the gap between their performance and two pruning-at-convergence methods, weight rewinding and magnitude pruning (Renda et al., 2020; Frankle et al., 2020).

In Figure 2 we evaluate ProsPr on this benchmark using ResNet-20 and VGG-16 on CIFAR-10, and ResNet-18 on Tiny-ImageNet. It can be seen that ProsPr reduces the performance gap, especially at higher sparsity levels, and in some cases exceeds the accuracy of pruning-after-convergence methods. Full results are also summarised in Appendix B.

This is a remarkable achievement: ProsPr is the first work to close the gap to methods that prune after training. Previous works that prune at the start have not been able to outperform methods that prune after training on any settings, including smaller datasets such as CIFAR-10 or Tiny-ImageNet. It is also important to note that other baselines that have comparable accuracies are all iterative methods. ProsPr is the only method that can do this in a single shot after *using only 3 steps* batch-sizes of 512

Table 1: Test accuracies of VGG-19 and ResNet-50 on ImageNet. First-Order ProsPR exceeds the results reported by de Jorge et al. (2021) in all configurations but one, where GraSP works best.

| Sparsity | VGG-19 | | | | ResNet-50 | | | |
| | 90% | | 95% | | 90% | | 95% | |
| Accuracy | Top-1 | Top-5 | Top-1 | Top-5 | Top-1 | Top-5 | Top-1 | Top-5 |
|---|---|---|---|---|---|---|---|---|
| Unpruned Baseline | 73.1 | 91.3 | — | — | 75.6 | 92.8 | — | — |
| ProsPr (ours) | **70.75** | **89.9** | 66.1 | 87.2 | **66.86** | **87.88** | **59.62** | **82.82** |
| FORCE | 70.2 | 89.5 | 65.8 | 86.8 | 64.9 | 86.5 | 59.0 | 82.3 |
| Iter-SNIP | 69.8 | 89.5 | 65.9 | 86.9 | 63.7 | 85.5 | 54.7 | 78.9 |
| GRASP-MB | 69.5 | 89.2 | **67.6** | **87.8** | 65.4 | 86.7 | 46.2 | 66.0 |
| SNIP-MB | 68.5 | 88.8 | 63.8 | 86.0 | 61.5 | 83.9 | 44.3 | 69.6 |
| Random | 64.2 | 86.0 | 56.6 | 81.0 | 64.6 | 86.0 | 57.2 | 80.8 |

in the inner-loop before computing the meta-gradients. In total, we only use 4 batches of data. We also do not do any averaging of scores by repeating the method multiple times.

The performance in these small datasets comes from the fact that ProsPr computes higher-order gradients. While there are other iterative methods that can work without any data, their effect is mostly a more graceful degradation at extreme pruning ratios as opposed to best accuracy at more practical sparsity levels. One example is SynFlow which is similar to FORCE but uses an all-one input tensor instead of samples from the training set (Tanaka et al., 2020).

## 4.2 RESULTS ON IMAGENET DATASET

To evaluate the performance of ProsPr on more difficult tasks we run experiments on the larger ImageNet dataset. Extending gradient-based pruning methods to this dataset poses several challenges.

**Number of classes** In synaptic-saliency methods, the mini batches must have enough examples from all classes in the dataset. Wang et al. (2020) recommend using class-balanced mini-batches sized ten times the number of classes. In datasets with few classes this is not an issue and even a single batch includes multiple examples per class. This is one reason why methods like SNIP work with a single batch, and why we kept the number of steps in ProsPr's inner loop fixed to only 3. ImageNet however has 1,000 classes, and using a single or a handful of small batches is inadequate. Previous methods such as FORCE, GraSP, or SynFlow avoid this problem by repeating the algorithm with new data batches and averaging the saliency scores. In ProsPr we instead increase the number of updates before computing the meta-gradients, ensuring they flow through enough data. Computing meta-gradients through many steps however poses new challenges.

**Gradient degradation** We start to see gradient stability issues when computing gradients over deep loops. Gradient degradation problems, i.e., vanishing and exploding gradients, have also been observed in other fields that use meta-gradients such as Meta-Learning. Many solutions have been proposed to stabilize gradients when the length of loop increases beyond 4 or 5 steps, although this remains an open area of research (Antoniou et al., 2019).

**Computation Complexity** For ImageNet we must make the inner loop hundreds of steps deep to achieve balanced data representation. In addition to stability issues, backpropagating through hundreds of steps is very compute intensive.

Therefore for our experiments on ImageNet we use the first-order approximation of ProsPr (Sec 3.2). We evaluate ProsPr using ResNet-50 and VGG-19 architectures and compare against state-of-the-art methods FORCE and Iter-SNIP introduced by de Jorge et al. (2021). We include multi-batch versions of SNIP and GraSP (SNIP-MB and GraSP-MB) to provide a fair comparison to iterative methods, which partially prune several times during training, in terms of the amount of data presented to the method. We use 1024 steps with a batch size of 256 (i.e. 262,144 samples) for ResNet-50. For VGG-19, a much larger model, and which requires more GPU memory we do 256 steps with batch size of 128. This is still far fewer samples than other methods. Force, for example, gradually prunes

in 60 steps, where each step involves computing and averaging scores over 40 batches of size 256, i.e. performing backpropagation 2400 times and showing 614,400 samples to the algorithm.

Table 1 shows our results compared to the baselines reported by de Jorge et al. (2021). First-order ProsPr exceeds previous results in all configurations except one, where it is outperformed by GraSP. Note the surprisingly good performance of random pruning of ResNets, which was also observed by de Jorge et al. (2021). This could be explained by the fact that VGG-19 is a much larger architecture with 143.6 million parameters, compared to 15.5 million in ResNet-50s. More specifically the final three dense layers of VGG-19 constitute 86% of its total prunable parameters. The convolution layers of VGG constitute only 14% of the prunable weights. Pruning methods are therefore able to keep more of the convolution weights and instead prune extensively from the over-parametrized dense layers. ResNet architectures on the hand have a single dense classifier at the end.

## 4.3 STRUCTURED PRUNING

We also evaluate ProsPr in the structured pruning setup where instead of pruning individual weights, entire channels (or columns of linear layers) are removed. This is a more restricted setup, however it offers memory savings and reduces the computational cost of training and inference.

Adopting ProsPr for structured pruning is as simple as changing the shape of the pruning mask $\mathbf{c}$ in Eq 3 to have one entry per channel (or column of the weight matrix). We evaluate our method against 3SP, a method that extends SNIP to structured pruning (van Amersfoort et al., 2020). Our results are summarized in Table 2 which show accuracy improvements in all scenarios. In Appendix C we also evaluate wall-clock improvements in training time as a result of structured pruning at initialization.

Table 2: Test accuracies for structured pruning using VGG-19 on CIFAR-10 and CIFAR-100. ProsPr achieves better accuracy in all configurations.

| Sparsity | Method | CIFAR-10 Acc (%) | CIFAR-100 Acc (%) |
|---|---|---|---|
| — | Unpruned Baseline | 93.6 | 72.5 |
| 80% | ProsPr (ours) | **93.61 ± 0.01** | **72.29 ± 0.11** |
| | 3SP | 93.4 ± 0.03 | 69.9 ± 0.14 |
| | 3SP + reinit | 93.4 ± 0.04 | 70.3 ± 0.16 |
| | 3SP + rescale | 93.3 ± 0.03 | 70.5 ± 0.13 |
| | Random | 92.0 ± 0.08 | 67.5 ± 0.16 |
| 90% | ProsPr (ours) | **93.64 ± 0.24** | **71.12 ± 0.26** |
| | 3SP | 93.1 ± 0.04 | 68.3 ± 0.12 |
| | 3SP + reinit | 93.0 ± 0.02 | 69.0 ± 0.08 |
| | 3SP + rescale | 93.0 ± 0.06 | 69.2 ± 0.11 |
| | Random | 90.4 ± 0.12 | 63.8 ± 0.13 |
| 95% | ProsPr (ours) | **93.32 ± 0.15** | **68.03 ± 0.38** |
| | 3SP | 92.5 ± 0.12 | 63.2 ± 0.52 |
| | 3SP + reinit | 92.6 ± 0.09 | 64.2 ± 0.35 |
| | 3SP + rescale | 92.5 ± 0.06 | 63.5 ± 0.63 |
| | Random | 89.0 ± 0.15 | 60.1 ± 0.29 |

## 4.4 NUMBER OF META STEPS

Finally, we evaluate ProsPr when using a varying number of meta steps, which gives insight into whether using meta-gradients is beneficial. We repeated experiments from Section 4.3 but this time we vary the depth of training steps between 0 and 3. The results in Table 3 show that the final accuracy consistently increases as we increase the depth of the training, showing the effectiveness of meta-gradients. We used the same data batch in all M training steps to isolate the effect of M, while in other experiments we use a new batch in every step.

In theory increasing the number of training steps should always help and match the ultimate objective (estimating the loss after many epochs of training) in the limit. However, in practice increasing the number of steps beyond 3 poses a lot of gradient stability issues (and is computationally expensive). These issues have been also identified in the meta-learning literature (Antoniou et al., 2019).

Table 3: Evaluating the effect of meta steps (M) on structured pruning performance of VGG-19

(a) 90% Sparsity

| M | CIFAR-10 Acc | CIFAR-100 Acc |
|---|---|---|
| 0 | 93.1% ± 0.04 | 68.3% ± 0.12 |
| 1 | 93.3% ± 0.12 | 68.8% ± 0.18 |
| 2 | 93.5% ± 0.21 | 69.8% ± 0.20 |
| 3 | 93.6% ± 0.19 | 71.0% ± 0.21 |

(b) 95% Sparsity

| M | CIFAR-10 Acc | CIFAR-100 Acc |
|---|---|---|
| 0 | 92.5% ± 0.12 | 63.20% ± 0.52 |
| 1 | 92.9% ± 0.31 | 64.87% ± 0.35 |
| 2 | 93.25% ± 0.24 | 67.12% ± 0.42 |
| 3 | 93.29% ± 0.29 | 67.98% ± 0.31 |

## 5 RELATED WORK

**Pruning at initialization**  Several works extend the approach proposed by Lee et al. (2018). de Jorge et al. (2021) evaluate SNIP objective in a loop in which pruned parameters still receive gradients and therefore have a chance to get *un*-pruned. The gradual pruning helps avoid the layer-collapse issue, and their method, known as FORCE, achieves better performance at extreme sparsity levels. Tanaka et al. (2020) provide theoretical justification for why iteratively pruning helps with the layer-collapse issue and propose a *data-free* version of the method where an all-one input tensor is used instead of real training data. Wang et al. (2020) propose an alternative criterion to minimizing changes in the loss and instead argue for preserving the gradient flow. Their method, GraSP, keeps weights that contribute most to the *norm* of the gradients. van Amersfoort et al. (2020) extends SNIP and GraSP to *structured* pruning to make training and inference *faster*. They further augment the scores by their compute cost to push the pruning decision towards more FLOPS reduction.

**Gradual pruning**  As discussed in Section 1, in existing methods the training step has been absent from the saliency computation step. As a workaround, many methods make their approaches *training-aware* by applying pruning gradually and interleaving it with training: Zhu & Gupta (2018) proposed an exponential schedule for pruning-during-training and Gale et al. (2019) showed its effectiveness in a broader range of tasks. Frankle & Carbin (2019) show that weight rewinding achieves better results when done in multiple prune-retrain steps. Lym et al. (2019) continuously apply structured pruning via group-lasso regularization while at the same time increasing batch sizes. You et al. (2020) find pruned architectures after a few epochs of training-and-pruning and monitoring a distance metric.

**Meta-Gradients**  Backpropagation through gradients, and its first-order approximation, is also used in model-agnostic meta-learning literature (Finn et al., 2017; Zintgraf et al., 2019) where the objective is to find a model that can be adapted to new data in a few training steps. Similar to our setup, the meta-loss captures the trainability of a model, but additionally, the meta-gradients are used to update the network's weights in a second loop. In self-supervised learning setting, Xiao et al. (2021) use meta-gradients to explicitly optimize a learn-to-generalize regularization term in nested meta-learning loops. Computing gradients-of-gradients is also used to regularize loss with a penalty on the gradients, for instance, to enforce Lipschitz continuity on the network (Gulrajani et al., 2017) or to control different norms of the gradients (Alizadeh et al., 2020).

## 6 DISCUSSION

Although pruning at initialization has the potential to greatly reduce the cost of training neural networks, existing methods have not lived up to their promise. We argue that this is, in part, because they do not account for the fact that the pruned network is going to be *trained* after it is pruned. We take this into account, using a saliency score that captures the effect of a pruning mask on the training procedure. As a result, our method is competitive not just with methods that prune before training, but also with methods that prune iteratively during training and those that prune after training. In principle, compressing neural networks at initialization has the potential to reduce energy and environmental costs of machine learning. Beyond our context, taking into account that methods which prune-at-convergence generally have to be fine-tuned, it is possible that our work could have further implications for these pruning methods as well (Molchanov et al., 2016; Wang et al., 2019).

ACKNOWLEDGMENTS

Milad Alizadeh is grateful for funding by the EPSRC (grant references EP/R512333/1) and Arm (via NPIF 2017 studentship). Shyam Tailor is supported by EPSRC grants EP/M50659X/1 and EP/S001530/1 (the MOA project) and the European Research Council via the REDIAL project (Grant Agreement ID: 805194). Luisa Zintgraf is supported by the 2017 Microsoft Research PhD Scholarship Program, and the 2020 Microsoft Research EMEA PhD Award. Joost van Amersfoort is grateful for funding by the EPSRC (grant reference EP/N509711/1) and Google-DeepMind. Sebastian Farquhar is supported by the EPSRC via the Centre for Doctoral Training in Cybersecurity at the University of Oxford as well as Christ Church, University of Oxford.

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

# A    EXPERIMENTAL SETUP

## A.1    ARCHITECTURE DETAILS

We use standard VGG and ResNet models provided by `torchvision` throughout this work where possible. The ResNet-20 model, which is not commonly evaluated, was implemented to match the version used by Frankle et al. (2021) so that we could compare using the benchmark supplied by this paper.

For smaller datasets, it is common to patch models defined for ImageNet. Specifically, for ResNets, we replace the first convolution with one $3 \times 3$ filter size, and stride 1; the first max-pooling layer is replaced with an identity operation. For VGG, we follow the convention used by works such as FORCE (de Jorge et al., 2021). We do not change any convolutional layers, but we change the classifier to use a single global average pooling layer, followed by a single fully-connected layer.

## A.2    TRAINING DETAILS

For CIFAR-10, CIFAR-100 and TinyImageNet we perform 3 meta-steps to calculate our saliency criteria. We train the resulting models for 200 epochs, with initial learning rate 0.1; we divide the learning rate by 10 at epochs 100 and 150. Weight decay was set to $5 \times 10^{-4}$. Batch size for CIFAR-10, CIFAR-100, and TinyImageNet was 256. For CIFAR-10 and CIFAR-100 we augment training data by applying random cropping ($32 \times 32$, padding 4), and horizontal flipping. For TinyImageNet we use the same procedure, with random cropping parameters set to $64 \times 64$, padding 4.

For ImageNet we train models for 100 epochs, with an initial learning rate of 0.1; we divide the learning rate by 10 at epochs 30, 60 and 90. Weight decay was set to $1 \times 10^{-4}$. Batch size was 256. We use the first order approximation to do pruning, and use 1024 steps for ResNet-50. For VGG-19 we use 2048 steps, but with batch size set to 128 (due to memory limitations, as our implementation only utilized a single GPU for meta-training). We apply random resizing, then crop the image to $224 \times 224$, with horizontal flipping.

## A.3    IMPLEMENTATIONS

In addition to our code, the reader may find it useful to reference the following repos from related work. Our experiments were performed using code derived from these implementations:

- `https://github.com/naver/force`
- `https://github.com/alecwangcq/GraSP`
- `https://github.com/facebookresearch/open_lth`
- `https://github.com/ganguli-lab/Synaptic-Flow`
- `https://github.com/mil-ad/snip`

# B    NUMBERS FROM FIGURE 2

Table 4: Numerical results for ResNet-20 on CIFAR-10

| Sparsity (%) | 20.0 | 36.0 | 48.8 | 59.0 | 67.2 | 73.8 | 79.0 | 83.2 | 86.6 | 89.3 | 91.4 | 93.1 | 94.5 | 95.6 | 96.5 | 97.2 | 97.7 | 98.2 |
|---|---|---|---|---|---|---|---|---|---|---|---|---|---|---|---|---|---|---|
| LTR after Training | 91.8±0.2 | 91.9±0.2 | 91.9±0.2 | 91.7±0.2 | 91.5±0.1 | 91.4±0.1 | 91.1±0.1 | 90.6±0.1 | 90.1±0.0 | 89.2±0.1 | 88.0±0.2 | 86.8±0.2 | 85.7±0.1 | 84.4±0.2 | 82.8±0.1 | 81.2±0.3 | 79.4±0.3 | 77.3±0.5 |
| Magnitude after Training | 92.2±0.3 | 92.0±0.2 | 92.0±0.2 | 91.7±0.1 | 91.5±0.2 | 91.3±0.2 | 91.1±0.2 | 90.7±0.2 | 90.2±0.2 | 89.4±0.2 | 88.7±0.2 | 87.7±0.2 | 86.5±0.2 | 85.2±0.2 | 83.5±0.3 | 81.9±0.3 | 80.4±0.2 | 77.7±0.4 |
| Magnitude at Initialization | 91.5±0.0 | 91.2±0.1 | 90.8±0.1 | 90.7±0.2 | 90.2±0.1 | 89.8±0.2 | 89.3±0.2 | 88.6±0.2 | 87.9±0.3 | 87.0±0.3 | 86.1±0.2 | 85.2±0.4 | 83.9±0.2 | 82.5±0.4 | 80.7±0.5 | 79.1±0.4 | 77.2±0.4 | 74.5±0.7 |
| SNIP | 91.8±0.2 | 91.2±0.3 | 90.9±0.1 | 90.7±0.1 | 90.1±0.2 | 89.7±0.3 | 89.0±0.2 | 88.5±0.3 | 87.7±0.2 | 87.2±0.4 | 85.8±0.1 | 84.7±0.5 | 83.8±0.3 | 82.5±0.4 | 80.9±0.2 | 79.1±0.2 | 77.3±0.2 | 74.0±0.5 |
| GraSP | 91.5±0.1 | 91.3±0.2 | 91.2±0.1 | 90.6±0.2 | 90.3±0.2 | 89.6±0.1 | 89.1±0.2 | 88.4±0.2 | 87.9±0.1 | 87.0±0.2 | 85.9±0.1 | 85.1±0.4 | 83.9±0.4 | 82.8±0.2 | 81.2±0.2 | 79.7±0.3 | 78.0±0.3 | 76.0±0.5 |
| SynFlow | 91.7±0.1 | 91.3±0.2 | 91.2±0.1 | 90.8±0.1 | 90.4±0.2 | 89.8±0.1 | 89.5±0.3 | 88.9±0.4 | 88.1±0.1 | 87.4±0.5 | 86.1±0.2 | 85.4±0.2 | 84.3±0.2 | 82.9±0.2 | 81.7±0.2 | 80.0±0.3 | 78.6±0.4 | 76.4±0.4 |
| Random | 91.6±0.2 | 91.2±0.2 | 90.8±0.3 | 90.5±0.2 | 89.8±0.2 | 89.0±0.4 | 88.4±0.2 | 87.5±0.3 | 86.6±0.2 | 85.6±0.3 | 84.3±0.4 | 83.1±0.4 | 81.6±0.3 | 79.6±0.4 | 74.2±6.4 | 64.7±9.7 | 56.9±8.5 | 43.7±12.5 |
| ProsPr | 92.3±0.1 | 92.1±0.0 | 91.7±0.2 | 91.5±0.1 | 91.0±0.2 | 90.5±0.0 | 90.1±0.1 | 89.6±0.2 | 88.5±0.5 | 87.8±0.1 | 86.9±0.3 | 85.5±0.6 | 84.3±0.2 | 83.0±0.9 | 80.8±0.5 | 79.6±0.7 | 77.0±0.8 | 74.2±0.3 |

Table 5: Numerical results for VGG-16 on CIFAR-10

| Sparsity (%) | 20.0 | 36.0 | 48.8 | 59.0 | 67.2 | 73.8 | 79.0 | 83.2 | 86.6 | 89.3 | 91.4 | 93.1 | 94.5 | 95.6 | 96.5 | 97.2 | 97.7 | 98.2 |
|---|---|---|---|---|---|---|---|---|---|---|---|---|---|---|---|---|---|---|
| LTR after Training | 93.5±0.1 | 93.6±0.1 | 93.6±0.1 | 93.6±0.1 | 93.8±0.1 | 93.6±0.1 | 93.6±0.1 | 93.8±0.1 | 93.8±0.1 | 93.7±0.1 | 93.7±0.1 | 93.8±0.1 | 93.5±0.2 | 93.4±0.1 | 93.2±0.1 | 93.0±0.2 | 92.7±0.1 | 92.1±0.4 |
| Magnitude after Training | 93.9±0.2 | 93.9±0.2 | 93.8±0.1 | 93.8±0.1 | 93.9±0.1 | 94.0±0.2 | 93.8±0.1 | 93.9±0.2 | 93.9±0.2 | 93.7±0.2 | 93.7±0.2 | 93.5±0.1 | 93.5±0.1 | 93.3±0.2 | 93.0±0.1 | 92.9±0.1 | 91.7±0.8 |
| Magnitude at Initialization | 93.6±0.0 | 93.4±0.2 | 93.3±0.1 | 93.2±0.2 | 93.3±0.3 | 93.0±0.1 | 93.1±0.1 | 92.9±0.1 | 92.9±0.2 | 92.7±0.1 | 92.5±0.2 | 92.3±0.1 | 92.2±0.2 | 92.0±0.1 | 91.8±0.2 | 91.5±0.1 | 91.3±0.3 | 90.9±0.2 |
| SNIP | 93.6±0.1 | 93.4±0.1 | 93.3±0.1 | 93.4±0.2 | 93.3±0.2 | 93.4±0.1 | 93.1±0.1 | 93.1±0.1 | 93.1±0.1 | 93.1±0.1 | 92.9±0.1 | 92.9±0.1 | 92.8±0.2 | 92.8±0.1 | 92.3±0.2 | 92.2±0.1 | 92.1±0.1 | 91.5±0.1 |
| GraSP | 93.5±0.1 | 93.4±0.2 | 93.5±0.0 | 93.3±0.1 | 93.2±0.2 | 93.3±0.2 | 93.4±0.1 | 93.2±0.1 | 93.0±0.3 | 93.0±0.3 | 92.7±0.2 | 92.8±0.1 | 92.4±0.1 | 92.4±0.1 | 92.2±0.1 | 91.9±0.1 | 91.6±0.2 | 91.5±0.0 | 91.2±0.2 |
| SynFlow | 93.6±0.2 | 93.6±0.1 | 93.5±0.1 | 93.4±0.1 | 93.4±0.2 | 93.5±0.2 | 93.2±0.1 | 93.2±0.1 | 93.1±0.1 | 92.9±0.1 | 92.7±0.2 | 92.5±0.1 | 92.3±0.1 | 92.0±0.1 | 91.8±0.3 | 91.3±0.1 | 91.0±0.2 | 90.6±0.2 |
| Random | 93.6±0.3 | 93.2±0.1 | 93.2±0.2 | 93.0±0.2 | 92.7±0.2 | 92.4±0.2 | 92.2±0.1 | 91.7±0.1 | 91.2±0.1 | 90.8±0.2 | 90.3±0.2 | 89.6±0.2 | 88.8±0.2 | 88.3±0.4 | 87.6±0.1 | 86.4±0.2 | 86.0±0.4 | 84.5±0.4 |
| ProsPr | 93.7±0.2 | 93.7±0.1 | 93.9±0.1 | 93.8±0.1 | 93.8±0.1 | 93.5±0.2 | 93.6±0.1 | 93.4±0.3 | 93.5±0.2 | 93.3±0.1 | 93.0±0.1 | 93.0±0.1 | 92.8±0.3 | 92.7±0.1 | 92.6±0.1 | 92.2±0.1 | 92.1±0.2 | 91.6±0.4 |

Table 6: Numerical results for ResNet-18 on TinyImageNet

| Sparsity (%) | 20.0 | 36.0 | 48.8 | 59.0 | 67.2 | 73.8 | 79.0 | 83.2 | 86.6 | 89.3 | 91.4 | 93.1 | 94.5 | 95.6 | 96.5 | 97.2 | 97.7 | 98.2 |
|---|---|---|---|---|---|---|---|---|---|---|---|---|---|---|---|---|---|---|
| LTR after Training | 51.7±0.2 | 51.4±0.3 | 51.5±0.4 | 52.1±0.4 | 51.8±0.4 | 52.0±0.1 | 52.0±0.1 | 52.0±0.2 | 52.1±0.3 | 52.0±0.2 | 52.4±0.2 | 51.8±0.4 | 51.8±0.6 | 51.4±0.4 | 50.9±0.2 | 49.3±0.7 | 48.3±0.7 | 46.0±0.3 |
| Magnitude after Training | 51.7±0.3 | 51.4±0.1 | 51.7±0.2 | 51.5±0.3 | 51.7±0.4 | 51.4±0.5 | 51.1±0.3 | 51.4±0.4 | 51.3±0.4 | 51.1±0.6 | 51.7±0.3 | 51.3±0.3 | 51.8±0.4 | 51.2±0.3 | 51.1±0.2 | 50.4±0.2 | 49.0±0.2 | 47.8±0.5 |
| Magnitude at Initialization | 51.0±0.3 | 51.2±0.3 | 51.0±0.2 | 50.5±0.5 | 50.6±0.3 | 50.0±0.3 | 50.3±0.2 | 50.3±0.3 | 50.0±0.1 | 49.8±0.5 | 49.0±0.1 | 48.3±0.3 | 47.2±0.2 | 46.2±0.2 | 44.4±0.5 | 42.2±0.1 | 40.8±0.4 | 38.1±0.6 |
| SNIP | 51.4±0.2 | 51.5±0.3 | 51.4±0.3 | 51.3±0.5 | 51.6±0.4 | 51.4±0.5 | 51.9±0.6 | 51.5±0.3 | 51.0±0.2 | 51.2±0.7 | 50.6±0.3 | 50.1±0.3 | 49.2±0.3 | 47.8±0.2 | 46.7±0.1 | 45.2±0.4 | 44.5±0.3 | 42.3±0.3 |
| GraSP | 49.8±0.4 | 49.1±0.3 | 49.5±0.2 | 49.5±0.4 | 49.2±0.1 | 49.5±0.2 | 48.7±0.1 | 49.0±0.5 | 48.8±0.4 | 48.3±0.1 | 48.2±0.1 | 47.7±0.2 | 46.5±0.1 | 45.5±0.7 | 44.9±0.2 | 44.1±1.0 | 42.9±0.5 | 41.0±0.1 |
| SynFlow | 51.8±0.3 | 51.6±0.3 | 51.7±0.7 | 51.8±0.2 | 51.3±0.4 | 51.3±0.4 | 51.5±0.2 | 51.0±0.4 | 50.2±0.4 | 50.4±0.3 | 49.1±0.0 | 48.0±0.5 | 46.7±0.7 | 45.6±0.0 | 44.0±0.2 | 42.2±0.3 | 40.0±0.1 | 38.2±0.5 |
| Random | 50.6±0.5 | 50.1±0.2 | 49.9±0.3 | 48.7±0.2 | 48.0±0.4 | 48.0±0.6 | 46.4±0.1 | 45.9±0.5 | 44.7±0.2 | 43.6±0.3 | 42.7±0.2 | 41.4±0.4 | 40.2±0.2 | 37.2±0.2 | 36.2±0.7 | 34.0±0.4 | 32.2±0.5 | 30.0±0.3 |
| ProsPr | 51.8±0.4 | 51.4±0.7 | 51.2±0.9 | 52.0±0.2 | 51.8±0.1 | 51.2±0.4 | 52.0±0.3 | 51.6±0.7 | 51.1±0.4 | 50.7±0.6 | 50.9±0.3 | 50.8±1.2 | 51.1±0.7 | 50.8±0.5 | 50.8±0.3 | 49.6±0.6 | 49.2±0.2 | 46.9±0.7 |

## C  WALL CLOCK TIME FOR STRUCTURE PRUNING AT INITIALIZATION

When pruning is done at convergence, the benefits of having a compressed model (in terms of *memory saving* and *speed-up*) can only be utilized at inference/deployment time. However, with pruning-at-initialization these benefits can be reaped during training as well. This is especially true in the case of structured pruning, where pruning results in weight and convolutional kernels with smaller dimensions (as opposed to unstructured pruning, where we end up with sparse weights with the original dimensions). This means that in addition to memory savings, training take fewer operations which speeds up training. To evaluate the benefits of training at initialization in terms of speed improvements we measured the wall-time training time on an NVIDIA RTX 2080 Ti GPU for the architectures used in Section 4.3 (an additionally on ImageNet dataset). The results in Table 7 show that structured pruning with ProsPr can significantly reduce the overall training time.

Table 7: Wall-time Training Speed-ups by Structured Pruning at Initialization

| Dataset | Epochs | Model | Unpruned | 80% pruned | 90% pruned | 95% pruned |
|---|---|---|---|---|---|---|
| CIFAR-100 | 200 | ResNet-18 | 83.9 mins | 60.76 mins | 54.8 mins | 46.6 mins |
| CIFAR-100 | 200 | VGG-19 | 50.3 mins | 45.8 mins | 39.93 mins | 38.8 mins |
| Tiny-ImageNet | 200 | ResNet-18 | 9.79 hours | 8 hours | 5.4 hours | 4.92 hours |
| Tiny-ImageNet | 200 | VGG-19 | 5.75 hours | 4.0 hours | 3.38 hours | 2.7 hours |
| ImageNet | 90 | ResNet-18 | 73.7 hours | 72.15 hours | 65.9 hours | 64.6 hours |

## D  RESULTS ON SEGMENTATION TASK

An interesting, albeit less common, application for pruning models is within the context of segmentation. In a recent paper Jeong et al. (2021) train and prune the U-Net (Ronneberger et al., 2015) architecture on two image datasets from the Cell Tracking Challenge (PhC-C2DH-U373 and DIC-C2DH-HeLa). They use the classic multi-step approach of gradually applying magnitude-pruning interleaved with fine-tuning stages. To evaluate the flexibility of our method we used meta-gradients at the beginning of training (on a randomly initialized U-Net), prune in a single shot, and train the network once for the same number of epochs (50). We kept the training set-up the same as the baseline by Jeong et al. (2021) (i.e., resizing images and segmentation maps to (256,256), setting aside 30% of training data for validation) and similarly aim to find the highest prune ratio that does not result in IOU degradation. We report intersection-over-union (IOU) metric for the two datasets in Tables 8 and 9:

Table 8: Mean-IOU on U373 validation

| Method | Prune Ratio | Mean IOU |
|---|---|---|
| Unpruned | - | 0.9371 |
| Jeong et al. | 95% | 0.9368 |
| ProsPr | 97% | 0.9369 |

Table 9: Mean-IOU on HeLa validation

| Method | Prune Ratio | Mean IOU |
|---|---|---|
| Unpruned | - | 0.7514 |
| Jeong et al. | 81.8% | 0.7411 |
| ProsPr | 90% | 0.7491 |

These results show that our method works as well (or better) compared to this compute-expensive baseline, in the sense that we can prune more parameters while keeping the IOU score the same.

# E    SELF-SUPERVISED INITIALIZATION

To evaluate the robustness and consistency of our method against non-random initialization we ran experiments using BYOL to learn representations from unlabeled samples (Grill et al., 2020). We used ResNet-18 as a backbone and trained for 1000 epochs with an embedding size of 64. Unlike the vanilla ResNet-18 architecture used in Section 4.3 we used the commonly-used modified version of ResNet-18 for smaller inputs (removing the first pooling layer and modifying the first convolutional layer to have kernel kernel size of 3, stride of 1, and padding size of 1). We then used this trained ResNet18 as the initialization for our meta-gradient pruning method. After the pruning step, all layers were trained as before until convergence. All training hyper-parameters were kept as before. The results (final test accuracies for 95% pruning) are summarized in Table 10.

Table 10: Comparing test accuracies (%) of ProsPr on randomly initialized ResNet-18 and initialization from self-supervised learning (BYOL)

| Dataset | Random Init | BYOL init |
|---|---|---|
| CIFAR-10 | 93.6 | 93.62 |
| CIFAR-100 | 73.2 | 74.02 |

These results show the robustness of our method for this particular self-supervised initialization. Starting from a learned representation can be challenging because these representations are much closer to weight values at convergence, and therefore the magnitude of their gradients is significantly smaller than randomly initialized weights. However, this is less of a problem for meta-gradients as their magnitude is still significant due to back-propagation through training steps. This can be seen in Figure 3 which shows the L2 norm of gradients of each layer of a BYOL-initialized ResNet-18 for meta-gradients compared to normal gradients. It can be seen that meta-gradients provide a stronger signal compared to normal gradients.

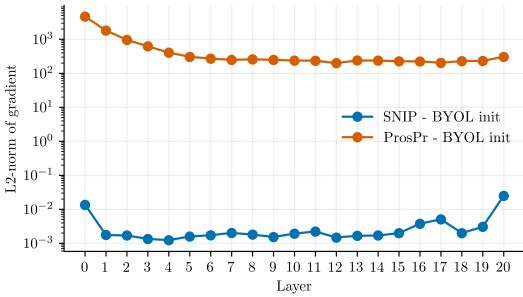

Figure 3: Comparing L2-norm of gradients and meta-gradients in BYOL-initialized ResNet-18

