# OpenReview forum: "Prospect Pruning: Finding Trainable Weights at Initialization using Meta-Gradients"
_ICLR.cc/2022/Conference — ICLR 2022 Poster_

### Official Review · Reviewer_MHq7 · 2021-10-17

**Correctness:** 3
**Technical Novelty And Significance:** 2
**Empirical Novelty And Significance:** 3
**Recommendation:** 6
**Confidence:** 4

**Main Review:**

Strength:
The overall method is well explained and easy to follow.

Weakness:
The performance of this work can be decoupled into two aspects:
1) If the meta-gradients w.r.t. masks ($\textbf{c}$ is vital, or single-step gradients (i.e. $M=1$) may also work.
2) If dropping small terms (Eq. 15 to 16) is indeed negligible.

I did not see ablation studies on these two aspects. Therefore I propose some further studies:
1) If the single-step gradient is used (no meta-gradients) to update the mask with the same total number of iterations (e.g. 1024 steps with a batch size of 256 for ResNet-50), what will be the pruning results? This is actually more GPU-memory efficient since no extra computational graphs are stored. Both "pruning by optimizing the mask" (this work) and magnitude pruning could be studied.
2) If we do not drop small terms in Eq. 15, what will be the pruning results? Smaller $M$ would be acceptable if storing computational graph is GPU-memory consuming.

In short, I think it is important to understand which part is vital to pruning: 1) optimizing the mask or magnitude pruning; 2) mata-gradients or just single-step gradients with more steps; 3) dropping or keeping the small terms.

**Summary Of The Paper:**

This work proposed a new efficient pruning method, by leveraging both loss sensitivity ("saliency score") during a few initial training steps. The authors leveraged meta-gradients with appropriate approximations to stabilize and speed up the pruning.

**Summary Of The Review:**

I generally think the novelty of this work is a bit trivial. Leveraging information from early training steps is a reasonable strategy, but is still straightforward.

---

> ### Author Response · Authors · 2021-11-17
> **Response to Reviewer MHq7**
>
> Thank you for your review, and the suggestion for additional ablation studies. We have done additional experiments to investigate the two aspects of why ProsPr works, as you suggested (see below). We will use these to clarify in the paper how each part of ProsPr (meta-gradients, multiple steps) is vital to good performance.
>
> ### Ablation: Effect of loop depth
> Below, we evaluate ProsPr when using a varying number of steps, which gives insight into whether using multiple steps is important.
> We repeated experiments from Section 4.3 but this time we increased the depth of training steps from 0 to 3. We observe that the final accuracy consistently increases as we increase the depth of the training, showing the effectiveness of meta-gradients:
>
> | Sparsity 	| Num Meta Steps (M) 	| CIFAR-10 Accuracy (%) 	| CIFAR-100 Accuracy 	|
> |----------	|--------------------	|-----------------------	|--------------------	|
> | 90%      	| 0                  	| 93.1% $\pm$ 0.04        	| 68.3% $\pm$ 0.12     	|
> | 90%      	| 1                  	| 93.3% $\pm$ 0.12        	| 68.8% $\pm$ 0.18     	|
> | 90%      	| 2                  	| 93.55% $\pm$ 0.21       	| 69.8% $\pm$ 0.20     	|
> | 90%      	| 3                  	| 93.61% $\pm$ 0.19       	| 71.09% $\pm$ 0.21    	|
>
> | Sparsity 	| Num Meta Steps (M) 	| CIFAR-10 Accuracy (%) 	| CIFAR-100 Accuracy 	|
> |----------	|--------------------	|-----------------------	|--------------------	|
> | 95%      	| 0                  	| 92.5% $\pm$ 0.12        	| 63.2% $\pm$ 0.52     	|
> | 95%      	| 1                  	| 92.9% $\pm$ 0.31        	| 64.87% $\pm$ 0.35    	|
> | 95%      	| 2                  	| 93.25% $\pm$ 0.24       	| 67.12% $\pm$ 0.42    	|
> | 95%      	| 3                  	| 93.29% $\pm$ 0.29       	| 67.98% $\pm$ 0.31    	|
>
> (Note that in these experiments we use the same data batch in all M training steps to isolate the effect of M, while in the paper we use a new batch in every step as that generally improves the performance. We experienced gradient stability issues when increasing above 4 which has been widely observed in meta-learning literature as well.)
>
> In theory increasing the number of training steps should always help and in the limit of large M, the match to the ultimate objective (estimating the loss after many epochs of training) would be exact. However, in practice increasing the number of steps beyond 3 or 4 poses a lot of gradient stability issues (and is computationally expensive).These issues have been also identified in the meta-learning literature and various solutions have been proposed to fix it. In our experiments above however, we can see the performance improving as we increase M from 0 to our chosen value, 3.
>
> ### Ablation: first-order approximation
>
> We’d like to emphasize that we only use the first-order approximation in the case of ImageNet dataset experiments where using the full meta-gradients is simply not possible due to the large number (1000) of classes in the dataset. For other datasets where we can compare full meta-gradients and first-order estimates we ran the same experiments above using the first-order approximation and observed negligible improvements compared to SNIP. We’ll update the paper with these results to highlight the importance of using full meta-gradients.
>
> Overall, these additional experiments show that show that:
>
> - Using multiple steps is beneficial, although with diminishing returns due to stability issues.
> - The higher-order terms (i.e., using meta-gradients rather than single-gradients with more steps) are necessary for good performance.
>
> We believe that these results, together with the experiments in the paper, give a better picture of why ProsPr works well: using meta-gradients over multiple steps is critical. The first-order approximation is still better than using only the point-estimate from SNIP, but should only be used if using the full ProsPr method is not possible due to memory constraints (like in our ImageNet experiments). From the SNIP paper [1] we also know that optimizing the mask, rather than magnitude pruning, is beneficial.
>
> We hope the above experiments and comments address your questions? We were a bit unsure about what you meant by:
>
> > If the single-step gradient is used (no meta-gradients) to update the mask with the same total number of iterations (e.g. 1024 steps wih a batch size of 256 for ResNet-50), what will be the pruning results?”, or if the above already answers this question.
>
> Do you mean to _update_ the mask at every timestep (which would mean after the first step, we can have values between 0 and 1 in the mask)? We hypothesise that this impacts the learning dynamics significantly, and that the final mask cannot be reliably used for pruning the initial weight vector. Or did you mean to compute the saliency mask using SNIP (i.e., single-step gradient given the current loss) at every timestep and then average?

---

> > ### Author Response · Authors · 2021-11-30
> > **Follow-Up**
> >
> > Dear reviewer, we hope you've had a chance to take a look at our response and revisions? We would really appreciate a reply as to whether our new experiments have addressed your questions, and whether they will allow you to update your score.

---

### Official Review · Reviewer_fSju · 2021-11-02

**Correctness:** 3
**Technical Novelty And Significance:** 3
**Empirical Novelty And Significance:** 3
**Recommendation:** 6
**Confidence:** 4

**Main Review:**

Strengths:

+ This paper is well-written.

+ the proposed method is easy to implement.

+ The experimental results support the claim that the proposed method achieves higher accuracy compared to existing pruning-at-init methods.

Weaknesses:

1. According to the proposed first-order approximation, it seems that the proposed method is limited to apply with the optimization method SGD. There is a variety of models that are trained with other optimization methods, like Adam, Adamw, and Adabound. It would be helpful to provide a discussion on the direction of how to adapt the proposed method to work with generic (or specific) optimization methods.

2. The first-order approximation (Eq. (17)) may not be well studied. The following questions are not clear: Is it sensitive to how $W_{init}$ is initialized (e.g., xavier_uniform, kaiming_normal, etc)? Is it robust to the sequence data? How does the number $M$ of initial training steps affect the accuracy and sparsity?

3. An important step to the first-order approximation is to drop the higher-order terms of Eq. (15), i.e., $\prod_{m=1}^{M} I - \alpha \nabla_{w_{m−1}}^{2} \mathcal{L} (w_{m−1}; \mathcal{D})$. A follow-up question is, what is the error bound (w.r.t. M) between the approximated meta-gradient and the ground-truth meta-gradient? For example, assume that the loss function meets the second-order necessary condition $\nabla_{w_{m−1}}^{2} \mathcal{L} (w_{m−1}; \mathcal{D}) \succeq 0$, as $M$ gets large enough, the operation $\prod_{m=1}^{M} I - \alpha \nabla_{w_{m−1}}^{2} \mathcal{L} (w_{m−1}; \mathcal{D})$ could lead to vanishing and exploding gradients. However, the approximated gradient $\nabla_{w_{M}} \mathcal{L}( w_{m−1}; \mathcal{D}) w_{init}$ seems to be still robust. How to interpret the discrepancy?

4. Figure 2 shows that pruning at convergence achieves a better trade-off between accuracy and sparsity than pruning at initialization. So it is not clear that the advantage(s) of the techniques of pruning at initialization over the ones of pruning at initialization. I looked into the introduction and the related work of pruning at initialization. I didn't find the points related to this. It would be better to explicitly discuss the comparison to make the motivation clearer. If I missed something, please point it out.

5. The algorithm presumes that $w_{init}$ is randomly initialized. It would be interesting to verify if the proposed method behaves consistently with $w_{init}$ that is learned from unlabeled samples, which is a common practice in self-supervised learning.

6. The related work of meta-gradients misses a related work, that is, meta-gradient in semi-supervised learning [r1].

7. This is not a critical comment, but I'd like to bring it to the discussion. Regarding the novelty, the proposed method relies on the saliency score (Eq. 1 and Eq. 11) defined in SNIP. This may make this work arguably look incremental. The authors present a discussion "our method, to be introduced in Section 3, also relies on computing the saliency scores for each element in the mask but uses a more sophisticated loss function to incorporate the notion of trainability into the objective." Still, I think the work is a bit weak in terms of the novelty of the methodology.

References:

[r1] Xiao, Taihong, Xin-Yu Zhang, Haolin Jia, Ming-Ming Cheng, and Ming-Hsuan Yang. "Semi-Supervised Learning with Meta-Gradient." In International Conference on Artificial Intelligence and Statistics, pp. 73-81. PMLR, 2021.

**Summary Of The Paper:**

This work studies the problem of pruning neural networks at initialization. It first identifies that the saliency score defined by the existing method SNIP has room for improvement. Specifically, the authors propose a method named prospect pruning to take into account the sequence of weight updates to determine the pruning mask. The experimental results on Tiny ImageNet and CIFAR show that the proposed method achieves better performance than existing methods of pruning at initialization.

**Summary Of The Review:**

This paper is well-written and provides detailed derivations. As a result, the proposed method has the potential to apply to a wide range of real-world applications. On the other hand, I think the proposed method may not be well studied and the experiment can be improved by adding the results of using $w_{init}$ learned from unlabeled data. Therefore, my initial recommendation is "marginally below the acceptance threshold". I will go over the review comments by other reviewers and the responses by the authors and adjust my recommendation accordingly.

---

> ### Author Response · Authors · 2021-11-17
> **Response to Reviewer fSju (Part 1)**
>
> Thank you for your thorough review! Below we address the main points you brought up. We hope you find this discussion and our new experiments convincing and consider increasing your score. If you still have any questions or concerns, please let us know.
>
> ### Initialization from Self-Supervised Learning (points 2 and 5)
> Verifying the robustness and consistency of our method against non-random initialization is an excellent suggestion. Following your comment, we ran additional experiments using BYOL [2] to learn representations from unlabeled samples in a ResNet18 backbone (trained 1000 epochs, embedding size of 64). We then used this trained ResNet18 as the initialization for our meta-gradient pruning method. After the pruning step, all layers were trained as before until convergence. All training hyper-parameters were kept as before. The results (final test accuracies for 95% pruning) are summarized below:
>
> |  Dataset  	| Random Init 	| BYOL init 	|
> |:---------:	|:-----------:	|:---------:	|
> | CIFAR-10  	| 93.6        	| 93.62     	|
> | CIFAR-100 	| 73.2        	| 74.02     	|
>
> These results show that our method is indeed robust to this particular self-supervised initialization and SSL init actually improves the performance slightly. However, your suggestions highlights an interesting practical scenario that could be challenging for previous gradient-based pruning methods and potentially could be used to set them apart. The reason why starting from a learned representation can be challenging is closely related to your other question regarding pruning-at-initialization vs. pruning-at-convergence. As these representations are much closer to weight at convergence, the magnitude of their gradients is significantly smaller than randomly initialized weight. However, this is less of a problem for meta-gradients as their magnitude is still significant due to backpropagation through several training steps. We’ll update the paper to include two new figures: (1) a train-accuracy plot that shows one initialization starts from a much better performance. (2) a plot showing L2 norm of gradients of each layer of a BYOL-initialized ResNet-18 for meta-gradients vs normal gradients that shows meta-gradients provide a stronger signal compared to gradients.
>
> We hope that these results address your questions.  If you have other experiments in this field that think make sense, we would love to hear about them.
>
> ### Missing Related Work: Meta-gradients for semi-supervised learning (point 6)
>
> Thank you for bringing the work by Zhang et al. [3] to our attention! While at first glance, the semi-supervised learning task may not seem very relevant to pruning, there are indeed many similarities between this particular work and our approach.
>
> By formulating SSL as a generalization problem, this paper uses meta-gradients to explicitly optimize for the objective they are interested in, rather than relying on proxies and assumptions (e.g. making assumptions the data manifold in the case of SSL). This is the main point we've also been trying to drive in our work, i.e., as pruning is always followed by training, we can capture its effect directly in meta-gradients.
>
> Similar to our work, they also discuss a similar first-order approximation and have a thorough analysis of convergence properties.
>
> We'll update the paper to cite this work. We will also study their convergence analysis more carefully to see if we can provide a similar analysis in future work.

---

> ### Author Response · Authors · 2021-11-17
> **Response to Reviewer fSju (Part 2)**
>
> ### Comparison of pruning-at-convergence and pruning-at-initialization (point 4)
> Deep learning pipelines (initialization, optimizers, architectures, etc.) have been tuned over the years to work best with over-parametrized models. However, as pruning results (and lottery-ticket-hypothesis) show, many tasks can achieve similar results with much smaller models. Thus, pruning at initialization can help us design more efficient pipelines.
>
> Compared to pruning at convergence, pruning at init is still in its early days, but as a recent succession of works (including this paper) shows, it is catching up in performance fast. Two aspects make pruning at initialization desirable:
>
> Pruning-at-convergence is computationally expensive. It is typically done either by estimating the Hessian of the loss, which is compute and memory intensive. Moreover, the pruned model often goes through many epochs of fine-tuning after pruning is done. While simpler criteria such as magnitude pruning have been shown to be effective, they're often applied in a multi-step iterative fashion, where each step includes many epochs of training in between. In contrast, our method is done after one meta-gradient step and works in a single-shot fashion.
>
> When pruning is done at convergence, the benefits of having a pruned model (in terms of "memory saving" and "speed-up") can only be enjoyed at inference/deployment time. However, with pruning-at-initialization these benefits can be reaped during training-time as well. This is especially true in the case of structured pruning, where pruning results in weight and conv kernels with _smaller dimensions_ (as opposed to unstructured pruning, where we end up with _sparse_ weights with the original dimensions). This means that in addition to memory savings, training iterations take up fewer operations which speed up training. We conducted additional experiments measuring the wall-time training time using an NVIDIA RTX 2080 ti for the architectures used in our experiments. It can be seen that structured pruning with ProsPr we can significantly reduce the overall training time:
>
> |    Dataset    	| Epochs 	|   Model   	|  Unpruned  	|  80% pruned 	| 90% pruned 	| 95% pruned 	|
> |:-------------:	|:------:	|:---------:	|:----------:	|:-----------:	|:----------:	|:----------:	|
> | CIFAR-100     	|   200  	| ResNet-18 	| 83.9 mins  	| 60.76 mins  	| 54.8 mins  	| 46.6 mins  	|
> | CIFAR-100     	|   200  	| VGG-19    	| 50.3 mins  	| 45.8 mins   	| 39.93 mins 	| 38.8 mins  	|
> | Tiny-ImageNet 	|   200  	| ResNet-18 	| 9.79 hours 	| 8 hours     	| 5.4 hours  	| 4.92 hours 	|
> | Tiny-ImageNet 	|   200  	| VGG-19    	| 5.75 hours 	| 4.0 hours   	| 3.38 hours 	| 2.7 hours  	|
> | ImageNet      	|   90   	| ResNet-18 	| 73.7 hours 	| 72.15 hours 	| 65.9 hours 	| 64.6 hours 	|
>
> **Potential future direction: meta-gradients for pruning-at-convergence**
>
> As discussed in the paper, when the model is at convergence, the gradient of loss is small (since that is what we're optimizing for during training), so the second-order Hessian dominates the Taylor expansion. Small gradients prohibit using gradient-based methods such as SNIP or FORCE, and pruning methods have instead focused on computing the Hessian. However, the meta-gradients at convergence are not necessarily small. In our additional experiments (see above), we observed this for self-super initialization. This makes us hopeful that our method can be used for pruning at convergence in the future. Compared to computing the Hessian, meta-gradients are significantly easier to implement thanks to autodiff implementations, and they include information that directly matches what we do in practice, i.e. prune and fine-tune.

---

> ### Author Response · Authors · 2021-11-17
> **Response to Reviewer fSju (Part 3)**
>
> ### Optimization (points 1, 2, 3)
>
> ProsPr can be used in combination with any differentiable optimization method (like Adam, Adamw, Adabound, etc.). The first-order approximation also works for these (under the same assumptions), except that the term which is dropped (r.h.s. of Equation 15) looks slightly different due to the use of things like momentum, etc. We will clarify in the paper; thanks for pointing this out.
>
> In additional experiments (see below), we evaluate ProsPr for a varying number of M steps, for the full model, and for the first-order approximation. We find that increasing the number of steps leads to increased performance (with diminishing return). We further find that the first-order approximation performs worse than the full ProsPr model. This points to the fact that the second-order terms contribute to the improved performance from ProsPr. It also shows that for small M, this does not suffer from vanishing/exploding gradients.
>
> | Sparsity 	| Num Meta Steps (M) 	| CIFAR-10 Accuracy (%) 	| CIFAR-100 Accuracy 	|
> |----------	|--------------------	|-----------------------	|--------------------	|
> | 90%      	| 0                  	| 93.1% $\pm$ 0.04        	| 68.3% $\pm$ 0.12     	|
> | 90%      	| 1                  	| 93.3% $\pm$ 0.12        	| 68.8% $\pm$ 0.18     	|
> | 90%      	| 2                  	| 93.55% $\pm$ 0.21       	| 69.8% $\pm$ 0.20     	|
> | 90%      	| 3                  	| 93.61% $\pm$ 0.19       	| 71.09% $\pm$ 0.21    	|
>
> | Sparsity 	| Num Meta Steps (M) 	| CIFAR-10 Accuracy (%) 	| CIFAR-100 Accuracy 	|
> |----------	|--------------------	|-----------------------	|--------------------	|
> | 95%      	| 0                  	| 92.5% $\pm$ 0.12        	| 63.2% $\pm$ 0.52     	|
> | 95%      	| 1                  	| 92.9% $\pm$ 0.31        	| 64.87% $\pm$ 0.35    	|
> | 95%      	| 2                  	| 93.25% $\pm$ 0.24       	| 67.12% $\pm$ 0.42    	|
> | 95%      	| 3                  	| 93.29% $\pm$ 0.29       	| 67.98% $\pm$ 0.31    	|
>
> We use the first-order approximation only in our ImageNet experiments, mainly due to memory constraints: on this complex task training takes longer, and hence a lot of update steps are necessary before computing the saliency score. We agree though that, memory constraints aside, the higher-order terms can vanish or explode as M increases. A recent preprint [1] has an interesting theoretical analysis on such types of failure modes, and an empirical study on meta-learned optimisers (which is a different problem setting than ours, but has similar properties) finds that as the number of steps increases, the loss landscape can become less smooth. A similar analysis, both theoretically and empirically, for ProsPr is an interesting direction for future work!
>
> ### Novelty (point 7)
>
> Thank you for the comment. We want to add that while our method uses SNIP objective as saliency criteria, ProsPr can be applied to any other differentiable gradient-based pruning criterion. We've chosen the prominent method SNIP because it's simple (estimates effects of perturbations using Taylor expansion) and well-studied.
>
> [1] Luke Metz, C. Daniel Freeman, Samuel S. Schoenholz, Tal Kachman - “Gradients are Not All You Need.” arXiv preprint arXiv:2111.05803 (2021).
>
> [2] Grill, Jean-Bastien, et al. "Bootstrap your own latent: A new approach to self-supervised learning." arXiv preprint arXiv:2006.07733 (2020).
>
> [3] Xiao, Taihong, Xin-Yu Zhang, Haolin Jia, Ming-Ming Cheng, and Ming-Hsuan Yang. "Semi-Supervised Learning with Meta-Gradient." In International Conference on Artificial Intelligence and Statistics, pp. 73-81. PMLR, 2021.

---

> ### Author Response · Authors · 2021-11-30
> **Follow-Up**
>
> Dear reviewer, we hope you've had a chance to take a look at our response and revisions? We would really appreciate a reply as to whether our new experiments have addressed your questions, and whether they will allow you to update your score.

---

> ### Comment · Reviewer_fSju · 2021-11-30
> **Feedback to the authors**
>
> Sorry for the late reply.
>
> Thank the authors for addressing my questions. I went over all the comments by the reviewers and the responses by the authors. The authors well addressed points 3, 4, 5, and 6. The analysis related to M steps in point 2 is clear.
>
> On the other hand, how existing initialization methods (e.g., xavier_uniform, kaiming_normal) affect the proposed method is still not clear. The response to point 1, namely the adaptivity to general optimization methods, is not convincing due to the lack of empirical evidence. Regarding point 7, the response may be a bit weak as the proposed method is based on the saliency score defined in SNIP.
>
> In summary, this work has merits for publication, especially exploring a new way of pruning at initialization to save training time, but the aforementioned points are unclear. Therefore, I would slightly raise my recommendation to 'marginally above the acceptance threshold', given the paper in its current shape and prospective changes summarized by the authors.

---

### Official Review · Reviewer_9cLN · 2021-11-02

**Correctness:** 4
**Technical Novelty And Significance:** 3
**Empirical Novelty And Significance:** 3
**Recommendation:** 5
**Confidence:** 3

**Main Review:**

[Strengths]
- The proposed ProsPr is a simple and effective pruning method that prunes weights at initialization. The authors propose to use meta-gradients to compute saliency scores when pruning weights.

- The higher-order temrs in meta-gradients can be further dropped (Equ. 16) such that saving the initial weights $w_{init}$ is enough when computing meta-gradients.

- Experimental results show that ProsPr achieves state-of-the-art pruning performance.


[Weaknesses]
- Writing needs improvement. E.g., "Many methods attempts...", "...the original, unpruned, model" and "Previous works that prune at the start have training have not been...".

- Although experimental results show that using estimation over several steps of gradient descent improves pruning performance, the connection between meta-gradients and motivation (i.e., the trainability of weights) is not strong and convincing enough.


**Summary Of The Paper:**

This work focuses on weight pruning at initialization. In this paper, the authors point out an important problem that the pruned subnetwork at initialization is going to be trained and previous prune-at-init methods ignore this fact. As a result, these prune-at-init methods ignore the trainability of weights. This paper proposes to use meta-gradients through the first few steps of optimization to determine which weights to prune. Experimental results show that ProsPr (this paper) achieves state-of-the-art pruning performance.

**Summary Of The Review:**

It is a good technical paper, but it requires better writing. Although the proposed ProsPr, which is implemented by meta-gradients, is effective, I think the novelty is limited.

---

> ### Author Response · Authors · 2021-11-17
> **Response to Reviewer 9cLN**
>
> Thank you for your review and suggestions! We have taken these into account and address your points below. We hope our explanation and additional experiments make you more confident of your review and lead to you increasing your score. Please let us know in case there are any concerns left.
>
> ### Connection between meta-gradients and motivation
>
> Thank you for pointing this out - we'll clarify this in writing and have performed additional ablation experiments to study this connection further. We outline both below and will include these in a revised version of the paper.
>
> Our ultimate goal is to pick a pruned model, A, which is more trainable than an alternative pruned model B. That means we want $\mathcal{L}(\hat{y}_A,y)$ to be lower than $\mathcal{L}(\hat{y}_B,y)$ at convergence (for a fixed pruning ratio). Finding the optimal pruning mask is generally infeasible since the training horizon is long (i.e., evaluation is costly) and the space of possible pruning masks is large. Therefore, approximate methods have to be used. Current prune-at-init methods, like SNIP, use the one-step gradient of the loss as a proxy for this, which amounts to just drawing a straight line in loss space from initialization. Instead, our approach uses the M-step meta-gradient. This picks a line in loss-space, which more closely predicts the eventual actual loss. This is because: 1) it smooths out over more steps; 2) it takes into account interactions between weights in the training dynamics. Crucially, in the limit of large M, the match to the ultimate objective is exact.
>
> We'd like to bring your attention to the additional experiments we did in response to your comments and Reviewer 4. To further study how well meta-gradients capture trainability of networks, we repeated experiments from Section 4.3, but this time we increased the depth of training steps from 0 to 3. We observe that the final accuracy consistently increases as we increase the depth of the training, showing the effectiveness of meta-gradients:
>
> | Sparsity 	| Num Meta Steps (M) 	| CIFAR-10 Accuracy (%) 	| CIFAR-100 Accuracy 	|
> |----------	|--------------------	|-----------------------	|--------------------	|
> | 90%      	| 0                  	| 93.1% $\pm$ 0.04        	| 68.3% $\pm$ 0.12     	|
> | 90%      	| 1                  	| 93.3% $\pm$ 0.12        	| 68.8% $\pm$ 0.18     	|
> | 90%      	| 2                  	| 93.55% $\pm$ 0.21       	| 69.8% $\pm$ 0.20     	|
> | 90%      	| 3                  	| 93.61% $\pm$ 0.19       	| 71.09% $\pm$ 0.21    	|
>
> | Sparsity 	| Num Meta Steps (M) 	| CIFAR-10 Accuracy (%) 	| CIFAR-100 Accuracy 	|
> |----------	|--------------------	|-----------------------	|--------------------	|
> | 95%      	| 0                  	| 92.5% $\pm$ 0.12        	| 63.2% $\pm$ 0.52     	|
> | 95%      	| 1                  	| 92.9% $\pm$ 0.31        	| 64.87% $\pm$ 0.35    	|
> | 95%      	| 2                  	| 93.25% $\pm$ 0.24       	| 67.12% $\pm$ 0.42    	|
> | 95%      	| 3                  	| 93.29% $\pm$ 0.29       	| 67.98% $\pm$ 0.31    	|
>
> Note that in these experiments we use the same data batch in all M training steps to isolate the effect of M, while in the paper we use a new batch in every step as that generally improves the performance. We experienced gradient stability issues when increasing above 4 which has been widely observed in meta-learning literature as well.
>
> ### Novelty
>
> While the method is arguably simple, we do believe there is novelty: to the best of our knowledge meta-gradients have not been used in the context of pruning, and are surprisingly effective. In addition to improved performance, we believe our results shed insight into a potential shortcoming of existing pruning methods, which do not take into account trainability, but instead are point estimates. We also want to emphasize that, while our method uses SNIP objective as saliency criteria, our proposed method ProsPr can be applied to any other gradient-based pruning criterion. We've chosen the prominent method SNIP because it's simple (estimates effects of perturbations using Taylor expansion) and well-studied.
>
> ### Writing
>
> Thanks for your suggestions to improve our writing; we will update the paper accordingly.

---

> > ### Author Response · Authors · 2021-11-30
> > **Follow-Up**
> >
> > Dear reviewer, we hope you've had a chance to take a look at our response and revisions? We would really appreciate a reply as to whether our new experiments have addressed your questions, and whether they will allow you to update your score.

---

### Official Review · Reviewer_D88v · 2021-11-06

**Correctness:** 4
**Technical Novelty And Significance:** 4
**Empirical Novelty And Significance:** 4
**Recommendation:** 8
**Confidence:** 4

**Main Review:**

1) Since the current methods are insufficient to enable this optimization and lead to a large degradation in model performance, the authors proposed a new method to identify a fundamental limitation, namely that their saliency criteria look at a single step at the start of training without considering the trainability of the network. The paper tackles an important problem.
2) The paper is well organized and easy to follow.
3) It would be better if the authors can validate their method on more tasks, such as human segmentation and image denoising, etc.

**Summary Of The Paper:**

This paper proposes Prospect Pruning (ProsPr) to handle the problem of short-sightedness of existing methods. The main idea is to use meta-gradients through the first few steps of optimization to determine which weights to prune.

**Summary Of The Review:**

See the main review.

---

> ### Author Response · Authors · 2021-11-17
> **Response to Reviewer D88v**
>
> Thank you so much for your encouraging review - it made us all very happy!
>
> We agree with you that the application of our method is not limited to classification tasks on vision datasets, and that validating it on other tasks is important. We kept the scope of experiments in this paper limited to classification to make comparison with well-established baselines easier. However, following your suggestion, we ran additional experiments on two segmentation tasks, which we will include in an updated version of the paper:
>
> **Validating ProsPr on segmentation tasks**
>
> An interesting, albeit less common, application for pruning models is within the context of segmentation. A recent paper by Jeong et al. (2021) [1] prunes the U-Net [2] architecture on two datasets from the Cell Tracking Challenge (PhC-C2DH-U373 and DIC-C2DH-HeLa). They use the classic multi-step approach of gradually applying magnitude-pruning interleaved with fine-tuning stages.
>
> To train ProsPr on this task, we use meta-gradients at the beginning of training (on a randomly initialized U-Net), prune in a single shot, and train the network once for the same number of epochs (50). We kept the training set-up the same as the baseline by Jeong et al. (i.e., resizing images and segmentation maps to (256,256), setting aside 30% of training data for validation).
>
> Jeong et al. (2021) aim to find the highest prune ratio that does not result in IOU degradation. We therefore report intersection-over-union (IOU) metric for the two datasets:
>
> | U373 Dataset | Prune Ratio | Mean IOU on Validation |
> |--------------|:-----------:|:----------------------:|
> | Unpruned     |      -      |          0.9371        |
> | Jeong et al. |     95%     |          0.9368        |
> | ProsPr       |     97%     |          0.9369        |
>
>
> | HeLa dataset | Prune Ratio | Mean IOU on Validation |
> |--------------|:-----------:|:----------------------:|
> | Unpruned     |      -      |         0.7514         |
> | Jeong et al. |    81.8%    |         0.7411         |
> | ProsPr       |     90%     |         0.7491         |
>
> These results show that our method ProsPr works as well (or better) compared to this compute-expensive baseline, in the sense that we can prune more parameters while keeping the IOU score the same. We believe these results highlight the flexibility of our approach and will include them in a revised version of the paper. Thank you for the suggestion!
>
> [1] Jeong, Taehee, et al. "Neural network pruning for biomedical image segmentation." Medical Imaging 2021: Image-Guided Procedures, Robotic Interventions, and Modeling. Vol. 11598. International Society for Optics and Photonics, 2021.
>
> [2] Ronneberger, Olaf, Philipp Fischer, and Thomas Brox. "U-net: Convolutional networks for biomedical image segmentation." International Conference on Medical image computing and computer-assisted intervention. Springer, Cham, 2015.

---

> > ### Author Response · Authors · 2021-11-30
> > **Follow-Up**
> >
> > Dear reviewer, we hope you've had a chance to take a look at our response and revisions? We would really appreciate a reply as to whether you believe our new experiments have strengthened the work, and whether they will allow you to update your score.

---

### Author Response · Authors · 2021-11-22
**Paper Revision**

We would like to thank all reviewers again for their time to evaluate our work, and their constructive feedback! We have taken care to address all concerns (see individual responses) and believe the reviewer’s suggestions have strengthened our paper. We have now updated the paper with the points the reviewers have asked for. Specifically:

**Writing**
- We polished the writing in the paper, and included the suggested improvements by reviewer 9cLN.
- We included a clearer note about the first-order approximation also being applicable to optimisers other than vanilla SGD (page 5, footnote 4). [Reviewer fSju]
- We made the connection between meta-gradients and our motivation clearer (Sec 3, page 3) and added an ablation experiment, see below. [Reviewer 9cLN]
- We included a sentence about the motivation for pruning-at-init over pruning-at-convergence in the introduction, and added an experiment on speed improvements, see below. [Reviewer fSju]
- We added the reference to “Meta-gradients for semi-supervised learning” by Zhang et al. to the related work section. [Reviewer fSju]

**New experimental results**
- We added experiments that highlight the importance of doing full meta-gradients, and what effect the meta-loop’s depth has on performance. See Section 4.4, page 8. [Reviewers 9cLN, fSju, MHq7]
- We added the training-time savings because of structured pruning at initialization, see Appendix C. [Reviewer fSju]
- We added the results for evaluating ProsPr on tasks other than classification, namely image segmentation results on U-Net. See Appendix D. [Reviewer D88v]
- We added additional experiments on the robustness to initialization: using ProsPr on networks with initialization from self-supervised BYOL. See Appendix E. (Reviewer fSju)

We hope that the reviewers appreciate our efforts in taking the feedback into account, and look forward to their responses.

---

### Decision · Program_Chairs · 2022-01-20

**Decision:**

Accept (Poster)

**Comment:**

This paper is proposed to address neural network pruning at initialization with the help of meta-gradients considering the high-order relations between loss and optimization of trainable sub-network. The paper is well organized and written with the clear logic. The discussions of related works, as well as their limitations, are comprehensive. To verify the proposed method, the authors have tested it on various benchmarks with different settings. Overall, the meta-learning idea for model pruning is relatively new, which may bring more inspirations to the community.